# Influence of Tree Species and Size on Bark Browsing by Large Wild Herbivores

**DOI:** 10.3390/plants11212925

**Published:** 2022-10-30

**Authors:** Bohdan Konôpka, Vladimír Šebeň, Jozef Pajtík, Lisa A. Shipley

**Affiliations:** 1National Forest Centre, Forest Research Institute, T. G. Masaryka 22, SK-960 01 Zvolen, Slovakia; 2Faculty of Forestry and Wood Sciences, Czech University of Life Sciences Prague, Kamýcká 129, CZ-165 000 Prague, Czech Republic; 3School of the Environment, Washington State University, Pullman, WA 99164-2812, USA

**Keywords:** *Populus tremula*, *Salix caprea*, *Sorbus aucuparia*, *Cervus elaphus*, stem bark removal, young forest stand

## Abstract

Although an important part of the ecosystem, large wild herbivores (LWH), especially red deer (*Cervus elaphus* L.), cause significant damage to economically valuable timber in forests of Central Europe. Recent work has demonstrated that less valuable softwood broadleaved trees can act as “biological control” that helps reduce bark browsing on more valuable trees in a mixed stand. To better understand the factors that influence how much bark area and mass are removed by LWH from these broadleaved trees, we took advantage of a novel “natural” experiment that occurred after a breach in a herbivory exclosure surrounding a 10-year old mixed broadleaved/conifer stand in the Western Carpathians in north-western Slovakia. We measured the area of old (up to 2 years previously) and new browsed patches on stems of common aspen (*Populus tremula* L.), common rowan (*Sorbus aucuparia* L.) and goat willow (*Salix caprea* L.), and their position along the vertical profile of the stem. The browsed bark area (cm^2^) was then converted to the bark mass (g) removed and the proportion of browsed bark to total bark (%) using conversion equations. Our models demonstrated that the amount of bark removed was influenced by tree species, stem diameter, age of browsing (old vs. new), and stem section along the vertical profile. LWH removed the most bark area from willow but the most bark mass from aspen because aspen had thicker bark than the other tree species. Bark browsing was greater on trees > 6 cm basal diameter. The distribution of bark browsing along the vertical profile was symmetrical (unimodal) with maximum intensity at 101–125 cm from the ground, which corresponds with the height most optimal for feeding by red deer. However, previous browsing in 2019 and 2020 caused new browsing on willow in 2021 to be focused in stem sections lower (51–75 cm) and higher (126–150 cm) than that optima. By quantifying browsing patterns and the amount of bark that is accessible to LWH for forage on the most attractive softwood broadleaved trees, our work will contribute to developing better methods for protecting commercially important species such as European beech (*Fagus sylvatica* L.) and Norway spruce (*Picea abies* L. Karst.) in areas of Central Europe that are greatly affected by increasing population density of LWH, especially red deer.

## 1. Introduction

Mammals, including large wild herbivores (LWH hereafter), are an important and inseparable part of forest ecosystems, contributing to ecological processes. For instance, they have significant effects on primary producers through nutrient cycling, energy flow, and the exertion of bottom-up and top-down processes [1]. LWH are often considered “ecosystem engineers” because their grazing and browsing can alter structure and functioning of ecosystems and biotic communities [2]. In fact, LWH affect ecosystems not only by consuming biomass through grazing and browsing but also through defecation, urination, and trampling [3]. LWH often cause significant changes in nutrient cycling, which might benefit or harm ecosystem conditions, depending on their population density, site fertility, and ecosystem productivity [3]. For example, LWH influence soil properties and biodiversity of flora and fauna, especially invertebrates, e.g., [1,4,5,6]. Optimal function of many ecosystems depends on diverse flora and fauna, especially native and co-evolved species [7,8,9]. When overabundant, LWH can excessively browse tree shoots, disturbing forest regeneration [10] and slowing down natural succession [11]. Intensive stem bark browsing by LWH can also decrease the amount [12] and quality of timber production [13] and can increase fungal infestation in wood, e.g., [14]. These changes can reduce economic gain from forests.

In Central Europe, red deer (*Cervus elaphus* L.) are the largest herbivores and cause the most damage to forest ecosystems [15,16]. Red deer consume a variety of plant species. Although they prefer grasses, a substantial amount of their diet, especially in winter, is also composed of foliage, branches, and the bark of trees and woody shrubs [17,18]. To maximize their intake of digestible energy and protein, red deer select the most nutritious plant species and plant parts with sufficient abundance that are available [19,20,21]. However, when population density is high, intraspecific competition forces animals to also consume lower quality forages [22].

In the Western Carpathians in north-western Slovakia, our previous study [23] showed that red deer browsed most frequently and intensely on soft broadleaved tree species rather than the economically important conifer, Norway spruce (*Picea abies* L. Karst). Although Norway spruce benefited from diminished competition after these broadleaved tree species were damaged by red deer, richness of tree species in the forest stand declined. Hence, the presence of economically unimportant softwood broadleaved tree species can shift browsing away from Norway spruce and other species valuable in the timber industry. At the same time, our research [23,24,25] and works from other European authors [26,27,28,29] showed that of all soft broadleaved species, common aspen (*Populus tremula* L.), common rowan (*Sorbus aucuparia* L.), and goat willow (*Salix caprea* L.) are the most attractive for red deer and other LWH. Evaluating how these species could provide biological protection for commercial tree species such as Norway spruce, European beech (*Fagus sylvatica* L.) and oaks (*Quercus* spp.) requires quantifying their forage potential, especially in form of stem bark biomass on these tree species, for red deer. 

Although several previous studies have indexed stem bark browsing as browsing rate among trees on a stand level or consumed bark area on a single tree level [30,31,32], no study has yet quantified forage potential on the individual tree or stand level because of difficulty modelling bark biomass available for browsing (edible potential). However, our recent work [33] developed bark mass models for four broadleaves tree species (*Acer pseudoplatanus* L., *Populus tremula* L., *Sorbus aucuparia* L., and *Salix caprea* L.). 

Therefore, we took advantage of a novel and fortuitous natural experiment in which previously protected 10-year old trees were suddenly exposed to LWH (mostly red deer) browsing after fence removal to quantify the mass of bark consumed on individual trees in a mixed broadleaved/conifer forest stand where softwood broadleaved species dominated. Because the starting point of our experiment represented entirely undamaged trees, we could examine browsing by LWH free from previous bark wounds. We measured bark mass removed by LWH from each tree, and examined the influence of tree species, stem diameter, age of browsing (old vs. new), and stem section along vertical profile on removal. We expected differences in browsing among tree species based on physical and chemical properties of bark and that stem browsing intensity would increase with tree size. We also assumed that the most intensive stem browsing would occur at the height most comfortable for LWH. Finally, we predicted that wounds on stems caused by LWH stripping in previous years would change the intensity and spatial distribution of bark browsing along the stem.

## 2. Results

The trees we measured ranged in height between 2.0 and 11.7 m, 1.6 and 5.7, 3.5, and 8.0 m for common aspen, common rowan, and goat willow, respectively (Table 1). Diameters d_0_ varied between 16.5 mm and 138.0 mm in common aspen, between 21.0 and 104.5 in common rowan, and between 20.5 and 1095 mm in goat willow. At the same time, while mean heights and diameters were very similar in common aspen and goat willow, those of common rowan were slightly smaller.

As much as 87.1% of all aspens had old browsing and 21.4% new browsing (Table 2). In rowan, 77.7% and 79.3% of old and new individuals, respectively, were browsed. The highest share of browsed trees was recorded in goat willow with 97.5% of old browsing and 87.5% of new browsing. As for browsed bark area along the stem profile, the largest values were in goat willow with 450 cm^2^ in old and 368 cm^2^ in new browsing. The measurements further showed (Table 2) that the highest percentage of browsed trees were in the stem sections of 101–125 cm (always old browsing, in aspen: 80.0%; in rowan: 72.1%; in willow: 89.6%), 76–100 cm (new browsing in aspen: 13.1% and in rowan: 57.0%), and 126–150 cm (new browsing in willow: 64.2%). At the same time, maximum values of browsed areas occurred most often for the same sections as were the percentage of browsed trees.

On the stem section level, all main effects (tree species, year of browsing, diameter class, and stem section) and all interactions significantly influenced the measures of bark browsing (BBA, BBM, and PBB) by LWH (Table 3). Likewise, along the entire stem profile, all main effects (tree species, year of browsing, and diameter class) and interactions had statistically significant effects on browsing indicators (BBA, BBM, and PBB; Table 4), except the effect of tree species to BBM (F value = 2.10; *p* = 0.124). 

More bark was browsed (as measured by BBA, BBM, and PBB) in old vs. new browsing (Figure 1). Although willow had the highest BBA (440 cm^2^/tree) and PBB (8.3%), aspen had the greatest BBM (43 g/tree). For new browsing, all browsing metrics were highest on willow (BBA = 320 cm^2^, BBM = 21 g, PBB = 5.7%) and lowest on aspen.

For nearly all diameter classes, tree species, and browsing metrics, trees in the largest diameter class (>6 cm) had been browsed to the greatest extent (Figure 2). The only exception was that the highest PBB in the intermediate diameter class (3–6 cm). The largest differences in metrics among diameter classes were found in rowan, which had a tenfold greater PBB between the <3 cm and >6 cm class.

Along the vertical profile of the tree stem, bark browsing had a symmetrical, unimodal distribution for each browsing metric and each species, with maximum browsing in the stem Section 101–125 cm from the ground (Figure 3) and negligible browsing above 200 cm. BBA and PBB were greatest for willow across all stem sections, but aspen had the largest values in BBM in the stem section starting at 101 cm. Maximum BBM, the most important metric for nutrition of LWH, were 5.2 g, 4.4 g, and 2.9 g in aspen, willow, and rowan, respectively. 

The distribution of browsing along the stem profile also differed between the years of browsing (Figure 4). Old browsing (years 2019 and 2020) demonstrated symmetric distributions with maximum values in the stem section of 101–125 cm. In contrast, new browsing (2021) showed a flatter relationship overall, with a bimodal distribution in willow.

When examining browsing distribution along the vertical stem profile among diameter classes (Figure 5), we found that the distribution of the thinnest trees (<3 cm) was left-sided, the moderately thick trees were generally symmetrical and the thickest trees showed a right-sided distribution. Thus, maximum values of PBB were in the stem sections of 51–75 cm, 101–125 cm, and 126–150 cm in the order from the thinnest to thickest diameter class. 

Finally, when examining the interaction between tree species, stem diameter, and the distribution of browsing along the stem profile, the thinnest trees for all species were slightly left-sided and in the thickest trees slightly right-sided (Figure 6). The effect of diameter class on browsing profiles was greater in rowan than willow and aspen.

## 3. Discussion

### 3.1. Main Factors Influencing Stem Bark Browsing

Our results indicated that stems of goat willow were the most intensively browsed by LWH, followed by common rowan and common aspen. The three tree species differed in all metrics of browsing (i.e., BBA, BBM, and PBB), except that willow and aspen had a similar BBM across a portion of the vertical stem profile. Although the area browsed on willow was higher than on aspen, aspen had a thicker bark (i.e., higher value of specific surface mass), so LWH were able to consume a great mass per unit area. Although each of these tree species is generally recognized as attractive for feeding by LWH [23,24,27,28,29], our results demonstrate a clear rank of selection. Our results indicate that goat willow is the most attractive species to LWH as manifested by the largest BBA and also PBB; at the same time, already, the small trees (diameter under 3 cm) in comparison with the other species were rather intensively browsed. The least browsed tree species (in both BBA and PBB) was aspen; however, it carries the most bark mass per area unit of stem surface accessible to LWH. 

Although all of these softwood broadleaved species were intensively browsed, Norway spruce and silver birch in the same research stand had almost no browsing damage on their stems (exact data not available). In contrast, many previous authors have observed serious damage on stems of Norway spruce in Central Europe caused by LWH, especially red deer [31,34,35,36]. These data were likely collected in spruce-dominated stands without significant proportions of attractive broadleaved trees. That provide a more desirable alternative food for LWH that would mitigate damage to spruces. On the other hand, not all softwood broadleaved trees are attractive to LWH. For example, stems of birches (*Betula* spp) are usually not browsed at all, most likely because the wood contains high concentrations of terpenoids, especially betulin [37], that deter feeding by herbivores.

Our results clearly showed that stem bark browsing varies with the vertical position along the stem profile. The most intensive stem bark browsing on willow, rowan, and aspen in our study occurred between 51 to 150 cm. Our findings support those reported in other locations and tree species. In a study from Scotland, the most wounds on stems caused by red deer browsing were between 51 to 150 cm [38], in Slovakia the highest percentage of stem bark browsing on rowan was found between 101 and 150 cm with upper occurrence in 200 cm [25], and Prien [39] indicated that red deer typically remove bark from the main stem at a height from 80 to 170 (occasionally up to 200 cm) above ground. These heights not surprisingly correspond with the height of a red deer’s shoulder [40].

Our results also demonstrated that stem thickness (expressed as three diameter classes) influenced intensity of stem bark browsing. For rowan and aspen, the most intensively browsed trees were in the diameter class above 6 cm, and the least browsed in the diameter class under 3 cm. A similar pattern, however, was not observed for goat willow. Because goat willow was the most attractive tree species; this pattern might reflect that LWH were willing to browse on stems that were more difficult to consume based on differences in stem flexibility. Stems that are too thin (diameter class under 3 cm) are very flexible and browsing might be difficult because LWH cannot press their teeth against the stem surface. This assumption is supported by the fact that different diameter classes also varied in the height along the vertical profile where the maximum browsing intensity was recorded (i.e., 51–75 cm, 101–125 cm, and 126–150 cm in the diameter class of <3.0, 3.0–6.0, and >6.0, respectively). Because bark browsing on very thin trees occurs at lower parts of stem than in thicker trees, this suggests that their upper parts are too flexible and might move when LWH try to browse on them.

A recent literature review [36] indicated that the presence of stem bark browsing on trees is related to stem diameter. Extent of stem bark browsing often decreases with stem diameter because small trees have more bark that is more digestible to LWH [13]. Similarly, studies from the Czech Republic on wound size reported a smaller stem diameter of wounded trees than for healthy trees [41]. On the other hand, wound size itself initially increases with stem diameter because clearly larger wounds can only be inflicted on large-diameter trees. In Austrian forests, the later decrease in wound size in older trees was explained in terms of wound recovery and the removal of the most heavily damaged trees during the stand thinning [36].

Our results showed that current year stem bark browsing might be influenced by previous browsing on the same stem. Specifically, intensive previous stem bark browsing on goat willows between 76 and 125 cm, which represented nearly 20% of bark surface, reduced the current year stem bark browsing and shifted the new bark browsing to stem sections located 51–76 cm and 126–150 cm. However, it is unclear whether the reduction in browsing on previously browsed trees was related to solely to the missing bark in the “attractive” stem sections, or other chemical and physical changes that limited or discouraged from browsing this stem section after previous damage.

Our measurements and calculations showed that although the most intensive browsing (and the largest area) by LWH occurred on goat willow and the least in aspen, the greatest amount of mass was removed from aspen. This discrepancy relates to different amount of bark mass per unit area that is mainly affected by bark thickness. Our bark models [33] showed that aspen has two to three times thicker bark than willow and rowan. Therefore, its specific surface mass (g/dm^2^) is more than twice that of goat willow and rowan, thus the efficiency of browsing and forage potential are also likely to be two to three-fold higher in aspen than in goat willow and rowan. Further, bark models [33] showed that specific surface mass decreases with distance from the stem base to the top and increases with tree size (diameter d_0_). Common aspens can have very thick and furrowed bark especially in the lower part of stem from rather young growth stages [42], which might prevent from browsing by LWH. However, the effects of tree characteristics on the relationship between stem bark browsing by LWH is a topic for further study.

### 3.2. Stem Bark Browsing and Biological Control

Our findings and those reported in other studies [20,21,43] suggested that presence of softwood broadleaved species can act as biological control against LWH browsing on economical tree species. Therefore, we recommend that these softwood broadleaved species (except birches in Central Europe) be retained within vulnerable forest stands during cleanings, thinning and tending. While these tree species are still attractive for stem bark browsing, foresters could leave enough individuals to provide sustainable forage and regeneration without undue competition with economically relevant tree species. Density of softwood broadleaved species would likely differ across sites and depend on management goals (e.g., the balance between productive and other ecosystem services of the forest). Although these softwood broadleaved species could be left scattered across the stand, foresters might find it more efficient to keep smaller groups at the edge of stands, along skidding lines and roads, or on the naturally occurring stand gaps. In more disturbed areas, these groupings could also contribute to better dispersion of large wild herbivores across the stands.

An alternative to maintaining and managing attractive softwood broadleaved species within economically important stands might be to manage these species on special plots with the primary goal of providing forage supply for LWH. Browse plots are normally about 0.25 ha, and although these plots would not produce wood for several decades, they would reduce damage to economically important stands grown in the vicinity [44]. These kind of browse plots would have the advantage of luring LWH from valuable forest stands. On the other hand, these specialized stands may concentrate large LWH around them. The most practical solution (besides in some cases increased hunting of LWH) would be to establish browse plots in areas with a naturally high concentration of LWH (e.g., stands near meadows and agricultural crops) and areas that have a high potential for natural regeneration of softwood broadleaved species. At the same time, browse plots should be placed as far as possible from high-quality commercial forest stands [45] because LWH, especially red deer, concentrate in areas with these sources of forage [46,47]. Foresters and game managers must assess whether the highest economic and ecological values are realized from managing soft broadleaved species that are valuable for browse through natural regeneration within a mixed commercial forest or by planting and managing browse plots on a case by case basis.

## 4. Materials and Methods

### 4.1. Site and Plot Characteristics

Our study area was part of a research-demonstration entity called “Husárik” [48]. The research-demonstration entity is located in the northernmost part of the Javorníky Mountains (Kysuce region; north-western Slovakia). The elevation of the site is about 700 m.a.s.l. The climate is characteristically cold and humid, with an average temperature nearly 16° C in July and less than −5° C in January, having a yearly average about 6° C. The duration of snow cover is approximately 100 days a year and annual precipitation about 1100 mm [49]. The bedrock of the Javorníky Mountains consists of clay-stones and shales, and the soil is prevailingly modally acidic, mostly clayey. The research-demonstration entity is located in the site to *Abieto-Fagetum* typology [50], with potential natural vegetation of European beech (*Fagus sylvatica* L.) and silver fir (*Abies alba* Mill.) forest. During the last two centuries, Norway spruce has become the dominant tree species in the most of the Kysuce region, similarly also in the research-demonstration entity [49]. Spruce dominated monocultures in the research-demonstration entity have been declining since about the beginning of the present millennium. This decline was likely caused by climatic extremes (e.g., the extremely dry growing period in 2003) and the subsequent bark beetle outbreaks. Therefore, the stands were harvested in 2010 as part of “incidental felling” focused on dead, dying, weakened or infested trees.

The research-demonstration entity is a part of nearly 3000-ha hunting ground including 2000 ha of forest and 1000 ha fields and grasslands. According to records from the local association, the hunting ground 15 red deer, 60 roe deer (*Capreolus capreolus* L.), 50 mouflons (*Ovis aries musimon* L.), and 50 fallow deer (*Dama dama* L.) were counted in early spring (before parturition) in 2010. Then, in 2020 a minimum of 35 red deer, 50 roe deer, 30 mouflon and 60 fallow deer were counted. Natural predators such as brown bears (*Ursus arctos*) Eurasian lynx (*Lynx lynx*) and grey wolves (*Canis lupus*) are rare, however, their population density was slightly increasing in the entire region during our study [49]. Thus, the predators likely have not significantly influenced the population density of LWH in the last two to three decades. 

In 2011, a variety of experimental plots were established in the research-demonstration entity to assess different reforestation and silvicultural approaches (e.g., natural, planted and their combinations; [51]). At the same time, a fence was built to protect a 5.12-ha section of the research-demonstration entity that hosted artificial regeneration experiments [48]. The fence was made of metal mesh (45 × 45 mm) 2.4 m high that completely prevented access by LWHs. For our sampling, we selected a portion of the exclosure near the fence outside of artificial restoration plots inside a sub-area covered by young forest originating exclusively from natural regeneration (seed dispersal from mature trees). Our previous studies [23] showed that stands within fenced areas were typical with a high growth rate of common aspen, common rowan, and goat willow, which were able to suppress other tree species, especially Norway spruce. The stand growing right outside the fenced area and exposed to LWH was dominated by Norway spruce, as typical for the region [23]. 

Approximately 7 years after the exclosure fence was built in 2011, some sections of the fence were damaged and a few LWH were occasionally seen inside the fenced area in the winter 2018-2019. Although the fence was repaired in 2019 (then in 2021), other damage in the fence happened in 2019, 2020 and 2021. Therefore, browsing on trees inside the exclosure was observed in 2019, 2020, and 2021. At that point, this fence experiment changed from a stand completely protected from herbivory to include a browsing experiment on softwood broadleaved species. The browsing was likely caused primarily by red deer and to a lesser extent fallow deer, but unlikely by roe deer and mouflon (see Červený et al. [15]). This “new” experiment allowed us to quantify stem bark browsing under high LWH pressure on attractive (palatable) tree species with no previous damage as of 2018 (when trees were 7–8 years old). Thus the location and intensity of browsing in 2019 and 2020 was not restricted by previous browsing, but 2021 browsing was influenced by missing bark patches from previous years. 

### 4.2. Tree Measurement, Calculations, and Analyses

In 2021, the sub-area selected for our experiment was about 10 years old and composed mostly of common rowan (55.4% of all individuals), Norway spruce (19.9%), goat willow (12.5%), silver birch (6.5%) and other broadleaved species with a substantial contribution of common aspen (together 6.7%; [23]). We found that although Norway spruce and silver birch had almost no damage by LWH, common aspen, common rowan, and goat willow were heavily browsed. The stand was very dense (i.e., 28,000 trees/ha) and the crowns were short from intense tree competition pressure, mostly located in the heights over 1.8 m. Therefore, although branch browsing in the canopy was rare, the stems were intensively stripped.

From this area, we randomly selected 120 individuals of common aspen, 120 individuals of goat willow, and 80 individuals (because they were less abundant) of common aspen. Each tree was measured for stem diameter at the base (d_0_) with digital callipers at a precision of ±0.1 mm and height with a Vertex 5 hypsometer (Haglöf Sweden AB, Bromma) with a precision of ±10 cm. Then, the vertical profile of stem was divided into 25 cm long sections delineated by white chalk. Diameters on lower and upper borders of the sections were measured with digital callipers at a precision of ±0.1 mm. Then, each debarked (browsed) area was measured with digital callipers (precision of ±0.1 mm). Because most of the debarked areas were rectangular, we measured by the height and width. When the debarked area extended across sections, the area was split between the sections. 

The total (pre-browsed) bark surface (*S_b_*, cm^2^) was calculated for individual stem sections using the formula for the surface of the truncated cone omitting areas of base and top circles:$$S_{b}=\pi \left(r_{1}+r_{2}\right)\sqrt{{\left(r_{1}-r_{2}\right)}^{2}+l_{s}{}^{2}}$$
where

*r*_1_ is a radius of the bottom end (cm);*r*_2_ is a radius of the top end (cm);*l_s_* is the length of the section (25 cm).

Then, total bark surface along the stem profile up to the height of 250 cm was calculated as a sum of surfaces in ten 25 cm long consecutive stem sections. 

Bark volume was calculated for individual stem sections by multiplying bark surface by bark thickness. Bark thickness was obtained from species-specific models (Appendix A; adapted from Konôpka et al. 2022). Finally, bark mass of a stem section was quantified as the product of bark volume and bark density (see also [33]). We quantified stem bark browsing using three metrics: browsed bark area (BBA; cm^2^), browsed bark mass (BBM; g), and proportion of browsed bark to the total bark amount (PBB, %). Areas of browsing on stem section level and entire stem level were converted to bark mass by using specific surface mass, which express quantity of bark mass per unit area [33]. PBB was calculated as a proportion of browsed bark mass to total (pre-browsed) bark mass based on both stem section level and entire stem profile level (considering maximum distance from the ground level of 250 cm). 

We then evaluated the effects of four variables on intensity of stem bark browsing, including: (i) tree species (common aspen, common rowan, or goat willow); (ii) age of browsing (old with dark patches (2019 and 2020) vs. new with fresh white patches (2021); (iii) stem diameter class (≤3 cm vs. 3–6 cm vs. ≥6 cm); and iv) stem section divided into 10 sections each 25 cm long, along the vertical profile of stem, starting from the ground level (0–25 cm) up to 250 cm (226–250 cm).

Finally, we used ANOVA to analyse all main effects (tree species, year of browsing, stem diameter class, and stem section) and interactions. Because all interactions were significant (all P’s < 0.05), we conducted a post hoc ANOVAS to examine subsets of the variables in relation to our specific predictions/hypotheses. We first examined browsing on individual stem sections (four-way ANOVA including tree species, year of browsing, diameter class of stem, and section), and then examined total browsing within entire stem (three-way ANOVA including tree species, year of browsing, and diameter class of stem). Fishers Least Significant Difference test was used to compare means among all main effects and interactions for BBA, BBM, and PBB, using an alpha level of 0.05.

## 5. Conclusions

Using previously constructed models of stem bark and measurements on browsed tree stems in a novel “natural” experiment, we quantified the area and the mass of bark consumed by LWH. Although the largest area of bark was removed from goat willow, the greatest mass of bark was removed from aspen because of its thicker bark. Thicker bark likely increases the efficiency of browsing and forage potential of trees. Stem bark browsing in two previous years limited further browsing on goat willow and the vertical profile of browsing shifted from symmetric (with a clear maximum in the optimal position for a browsing animal) to a bimodal pattern. Our findings also indicate that already browsed trees might have decreasing attractiveness or limited potential for continuous foraging by LWH. These findings contribute to a better understanding of how economically indifferent softwood broadleaved species such as aspen, rowan, and willow provide biological protection of commercial tree species. 

As stands mature, the importance of softwood broadleaved species for browsing is likely diminished and becomes less useful for biological control of damage in commercial forests. Although as softwood broadleaved tree species mature and potentially lose importance in biological control to LWH, their occurrence is often beneficial in terms of enhancing biodiversity in forest ecosystems. At the same time, their presence in mature mixed forest stands would serve as a source of further reproduction, thus creating conditions for prospective coexistence of forest trees and LWH. Therefore, retaining a component of softwood broadleaved tree species in mixed forest stands might be an important precondition for successful forestry carried out under high LWH population density. Practically, softwood broadleaved species have about half the rotation period (nearly 50 years) than the most commercial species (90–110 years). Therefore, they might be harvested within a regular tending harvest in approximately middle of the life cycle of the commercial tree species. In such a case, harvested softwood broadleaved trees species together with individuals of commercial species (low-quality stems, damaged or infected, and therefore selected for tending harvest) might be used especially as a fuel or for energy production.

Finally, previous studies have shown that both commercially important and commercially unimportant trees create a suitable environment (e.g., thermal and security cover) and winter/spring forage for LWH [15]. Thus, addressing the interests of both the forestry and game management sectors requires managing for a sustainable population of LWH that minimizes damage to economically important forest stands [35] while providing adequate hunting opportunities [52].

## Figures and Tables

**Figure 1 plants-11-02925-f001:**
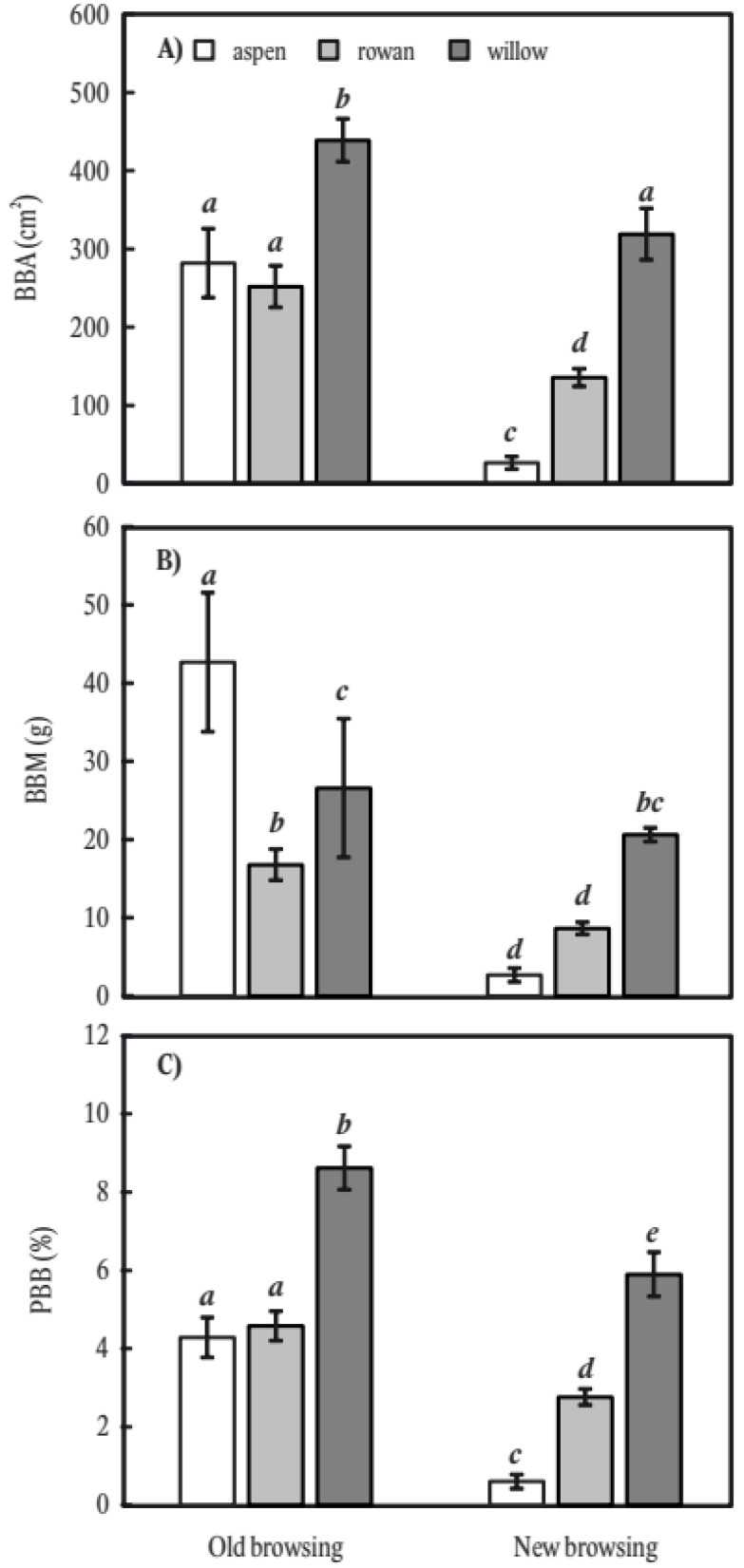
Stem bark browsing on common aspen (*Populus tremula* L.), common rowan (*Sorbus aucuparia* L.), and goat willow (*Salix caprea* L.) considering old (2019 and 2020) and new (2021) wounds. The stem browsing is expressed on the basis of area (**A**), mass (**B**) and the proportion of browsed bark mass to full bark mass potential (**C**). The error bars indicate standard errors. The different letters above the error bars indicate significant differences (three-way ANOVA and LSD test with alpha-level of 0.05).

**Figure 2 plants-11-02925-f002:**
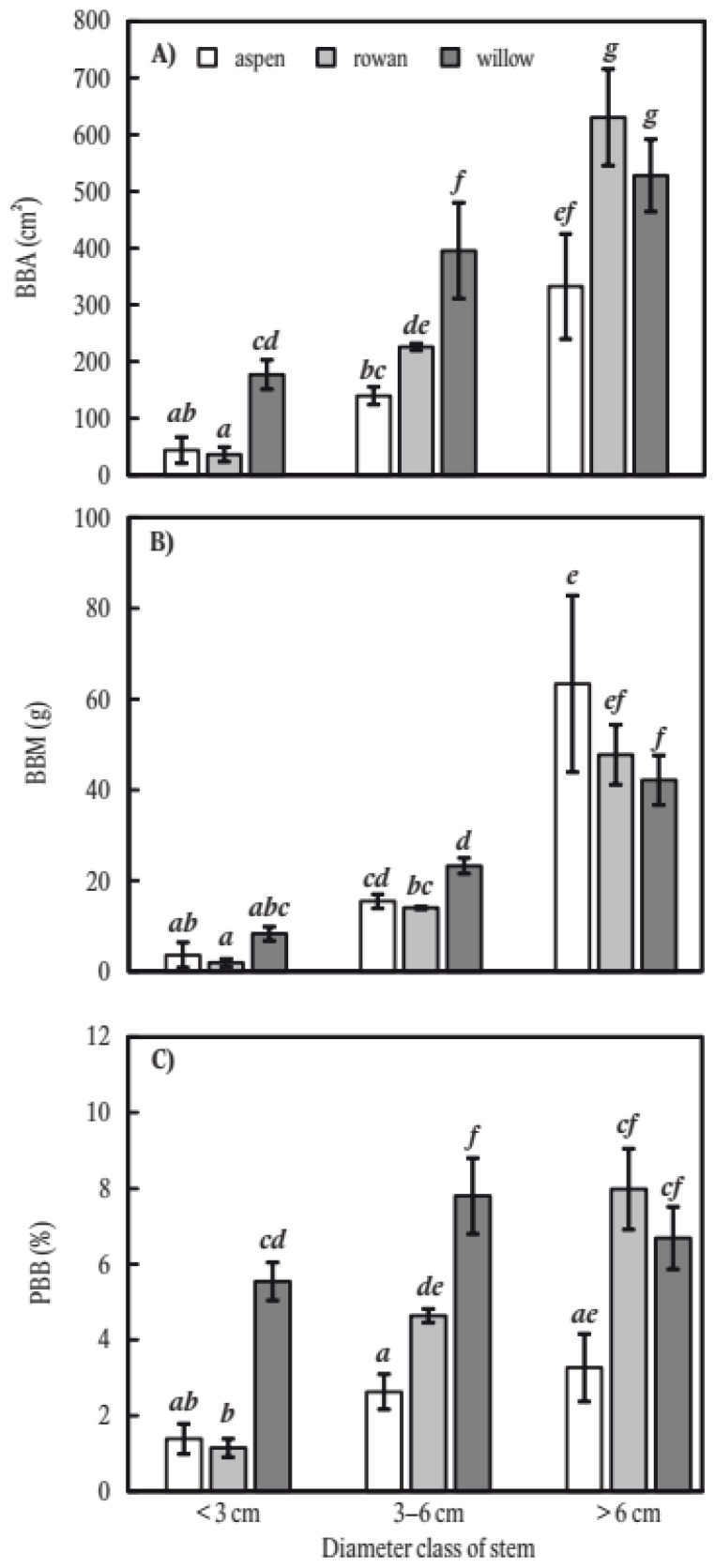
Stem bark browsing on common aspen (*Populus tremula* L.), common rowan (*Sorbus aucuparia* L.), and goat willow (*Salix caprea* L.) considering diameter classes of stems. The stem browsing is expressed on the basis of area (**A**), mass (**B**), and the proportion of browsed bark mass to full bark mass potential (**C**). The error bars indicate standard errors. The different letters above the error bars indicate significant differences (three-way ANOVA and LSD test with alpha-level of 0.05).

**Figure 3 plants-11-02925-f003:**
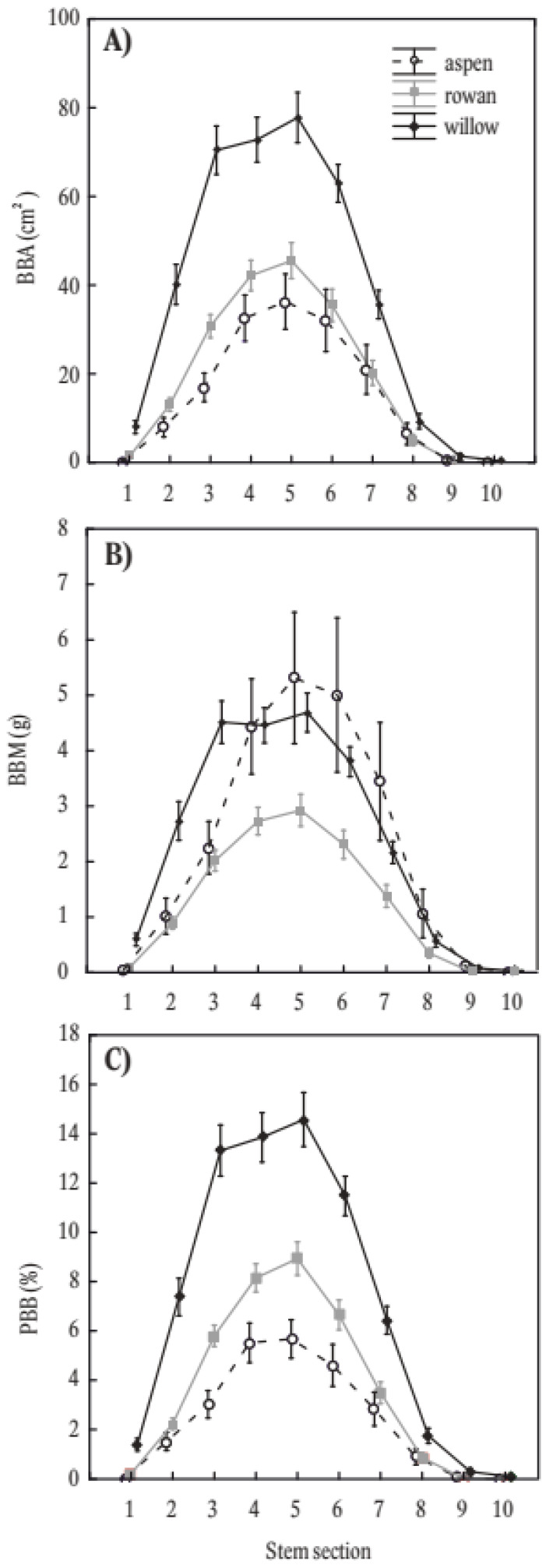
Stem bark browsing on common aspen (*Populus tremula* L.), common rowan (*Sorbus aucuparia* L.), and goat willow (*Salix caprea* L.) considering the stem sections. Stem browsing is expressed on the basis of area (**A**), mass (**B**) and the proportion of browsed bark mass to full bark mass potential (**C**). The error bars indicate standard errors. The stem sections are coded as 1: 0–25 cm, 2: 26–50 cm, 3: 51–75 cm, 4: 76–100 cm, 5: 101–125 cm, 6: 126–150 cm, 7: 151–175 cm, 8: 176–200 cm, 9: 2: 201–225 cm, and 10: 226–250 cm from the ground level.

**Figure 4 plants-11-02925-f004:**
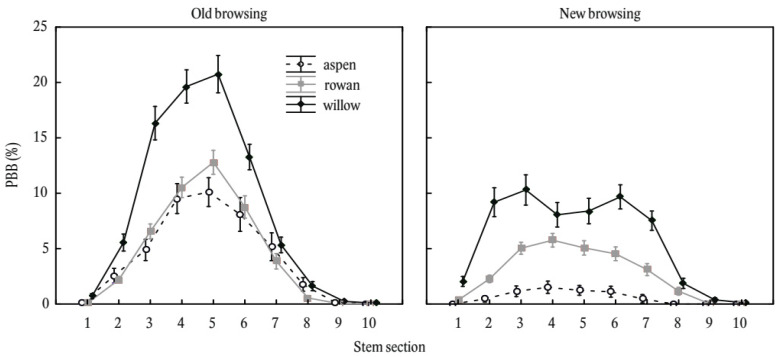
Stem bark browsing on common aspen (*Populus tremula* L.), common rowan (*Sorbus aucuparia* L.), and goat willow (*Salix caprea* L.) considering old (2019 and 2020) and new (2021) wounds by stem section. Stem browsing is expressed on relative basis (i.e., proportion of browsed bark mass to full bark mass potential). The error bars indicate standard errors. The stem sections are coded as 1: 0–25 cm, 2: 26–50 cm, 3: 51–75 cm, 4: 76–100 cm, 5: 101–125 cm, 6: 126–150 cm, 7: 151–175 cm, 8: 176–200 cm, 9: 2: 201–225 cm, and 10: 226–250 cm from the ground level.

**Figure 5 plants-11-02925-f005:**
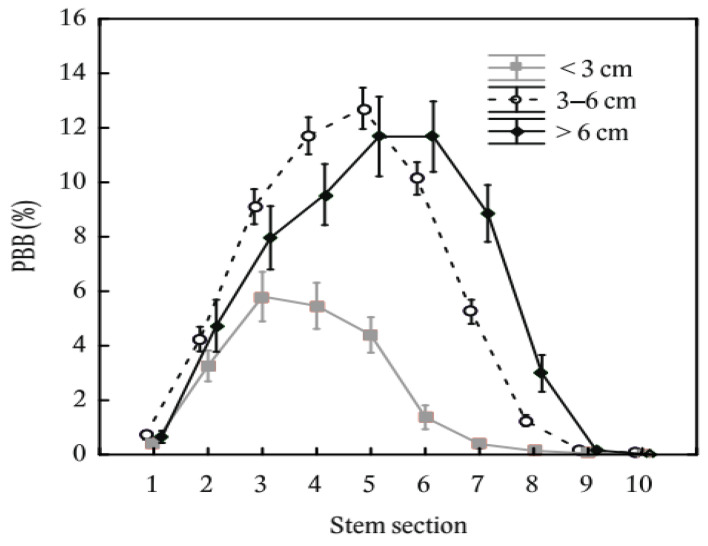
Stem bark browsing on common aspen (*Populus tremula* L.), common rowan (*Sorbus aucuparia* L.), and goat willow (*Salix caprea* L.) together considering stem sections and diameter classes of stems. The stem browsing is expressed on a relative basis (i.e., proportion of browsed bark mass to full bark mass potential). The error bars indicate standard errors. The stem sections are coded as 1: 0–25 cm, 2: 26–50 cm, 3: 51–75 cm, 4: 76–100 cm, 5: 101–125 cm, 6: 126–150 cm, 7: 151–175 cm, 8: 176–200 cm, 9: 2: 201–225 cm, and 10: 226–250 cm from the ground level.

**Figure 6 plants-11-02925-f006:**
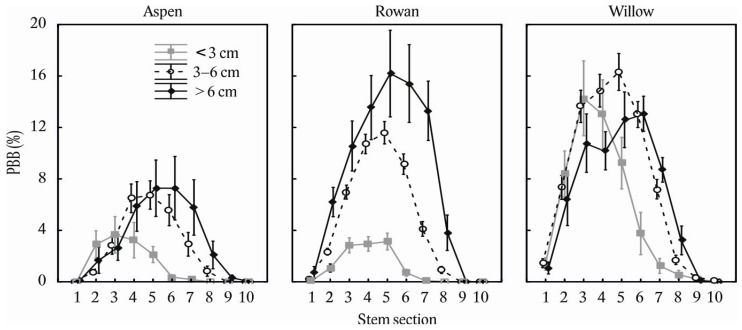
Stem bark browsing on common aspen (*Populus tremula* L.), common rowan (*Sorbus aucuparia* L.), and goat willow (*Salix caprea* L.) considering diameter classes of stems and stem sections. Stem browsing is expressed on relative basis (i.e., proportion of browsed bark mass to full bark mass potential). The error bars indicate standard errors. The stem sections are coded as 1: 0–25 cm, 2: 26–50 cm, 3: 51–75 cm, 4: 76–100 cm, 5: 101–125 cm, 6: 126–150 cm, 7: 151–175 cm, 8: 176–200 cm, 9: 2: 201–225 cm, and 10: 226–250 cm from the ground level.

**Table 1 plants-11-02925-t001:** Descriptive statistics representing the height (h, m) and stem base diameter (d_0_, mm) of common aspen (*Populus tremula* L.), common rowan (*Sorbus aucuparia* L.) and goat willow (*Salix caprea* L.), in the Husárik research-demonstration entity in the Kysuce region; north-western Slovakia.

Tree Species	Characteristcs	Mean	S.D.	Minimum	Maximum	25th Percentil	75th Percentil
Common aspen	Tree height	6.1	1.8	2.0	11.7	4.7	7.4
	Diameter d_0_	58.1	28.4	16.5	138.0	36.0	70.0
Common rowan	Tree height	4.7	0.6	1.6	5.7	3.3	4.1
	Diameter d_0_	49.4	16.5	21.0	104.5	38.0	55.0
Goat willow	Tree height	6.4	1.0	3.5	8.0	5.8	7.1
	Diameter d_0_	57.0	17.5	20.5	109.5	45.0	65.5

**Table 2 plants-11-02925-t002:** Mean share of browsed trees on total number of measured trees (%) and mean browsed area (cm^2^, in brackets) calculated for the set of browsed trees separately for old (years 2019 and 2020) and new (year 2021) bark damage by large wild herbivores by stem section on common aspen (*Populus tremula* L.), common rowan (*Sorbus aucuparia* L.) and goat willow (*Salix caprea* L.).

Stem Section (cm)	Common Aspen		Common Rowan		Goat Willow	
	Old Browsing	New Browsing	Old Browsing	New Browsing	Old Browsing	New Browsing
0–25	4.3 (12.2)	0	5.0 (9.2)	11.6 (18.9)	14.2 (27.9)	27.5 (43.9)
26–50	37.1 (38.9)	5.7 (25.6)	46.3 (27.9)	42.1 (31.4)	57.5 (49.0)	51.7 (101.0)
51–75	58.6 (49.9)	10.0 (47.1)	62.8 (58.3)	56.2 (44.4)	80.8 (102.0)	55.0 (106.1)
76–100	72.9 (80.7)	13.1 (49.5)	71.1 (80.2)	57.0 (48.0)	89.2 (113.6)	53.3 (82.9)
101–125	80.0 (84.5)	12.9 (38.6)	72.1 (97.3)	48.9 (45.7)	89.6 (123.6)	56.7 (80.1)
126–150	57.1 (101.7)	10.0 (60.2)	54.5 (90.7)	47.9 (44.8)	80.0 (91.4)	64.2 (82.2)
151–175	34.3 (113.9)	4.3 (68.2)	31.4 (74.1)	33.9 (50.7)	49.2 (61.3)	57.5 (71.5)
176–200	14.3 (90.3)	0	9.9 (29.2)	12.4 (58.2)	19.2 (43.2)	20.0 (51.8)
201–225	4.3 (30)	0	0.8 (13.8)	0	4.2 (23.1)	4.2 (46.3)
226–250	0	0	0.8 (7.4)	0	1.7 (18.4)	0.8 (59.2)
Full stem profile	87.1 (323.5)	21.4 (123.4)	77.7 (174.4)	79.3 (320.8)	97.5 (450.2)	87.5 (368.2)

**Table 3 plants-11-02925-t003:** Four-way ANOVA evaluating the influences of tree species (TS), year of browsing by large wild herbivores (YB), diameter class (DC), and stem section (SS) on the browsed bark area (BBA), browsed bark mass (BBM), and browsed bark mass to total bark mass (PBB) caused by large wild herbivores. Tree species were represented by common aspen (*Populus tremula* L.), common rowan (*Sorbus aucuparia* L.), and goat willow (*Salix caprea* L.).

Factors vs. Indicators	BBA			BBM			PBB		
	Df	F Value	*p* Value	Df	F Value	*p* Value	Df	F Value	*p* Value
TS	2	74.88	<0.001	2	6.79	<0.001	2	95.57	<0.001
YB	1	233.70	<0.001	1	312.76	<0.001	1	142.31	<0.001
DC	2	246.33	<0.001	2	311.89	<0.001	2	68.07	<0.001
SS	9	121.58	<0.001	9	90.05	<0.001	9	99.10	<0.001
TSxYB	2	37.41	<0.001	2	114.82	<0.001	2	12.21	<0.001
TSxDC	4	17.17	<0.001	4	14.52	<0.001	4	15.21	<0.001
TSxSS	18	6.65	<0.001	18	4.35	<0.001	18	7.96	<0.001
YBxDC	2	63.57	<0.001	2	118.66	<0.001	2	26.84	<0.001
YBxSS	9	43.38	<0.001	9	44.48	<0.001	9	26.92	<0.001
DCxSS	18	19.63	<0.001	18	23.13	<0.001	18	10.43	<0.001
TSxYBxDC	4	34.58	<0.001	4	68.97	<0.001	4	15.75	<0.001
TSxYBxSS	18	3.29	<0.001	18	7.6	<0.001	18	1.86	0.015
TSxDCxSS	36	1.44	0.043	36	2.69	<0.001	36	1.48	0.032
YBxDCxSS	18	13.11	<0.001	18	18.7	<0.001	18	6.22	<0.001
TSxYBxDCxSS	36	3.99	<0.001	36	5.29	<0.001	36	2.51	<0.001

**Table 4 plants-11-02925-t004:** Three-way ANOVA evaluating the influence of tree species (TS), year of browsing by large wild herbivores (YB), and diameter class (DC); browsed bark area (BBA), browsed bark mass (BBM), and browsed bark mass; and total bark mass (PBB) caused by large wild herbivores on the tree level.. Tree species were represented by common aspen (*Populus tremula* L.), common rowan (*Sorbus aucuparia* L.), and goat willow (*Salix caprea* L.).

Factors vs. Indicators	BBA			BBM			PBB		
	Df	F Value	*p* Value	Df	F Value	*p* Value	Df	F Value	*p* Value
TS	2	24.00	<0.001	2	2.10	0.124	2	33.39	<0.001
YB	1	74.91	<0.001	1	96.64	<0.001	1	46.50	<0.001
DC	2	78.96	<0.001	2	96.37	<0.001	2	19.90	<0.001
TSxYB	2	11.10	<0.001	2	35.48	<0.001	2	3.91	0.021
TSxDC	4	5.50	<0.001	4	4.49	0.001	4	5.21	<0.001
YBxDC	2	20.38	<0.001	2	36.67	<0.001	2	8.46	<0.001
TSxYBxDC	4	11.9	<0.001	4	21.31	<0.001	4	5.39	<0.001

## Data Availability

Not applicable.

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
