# Peer review of "Influence of Tree Species and Size on Bark Browsing by Large Wild Herbivores"

_plants, 2022, doi:10.3390/plants11212925_

Round 1
Reviewer 1 Report
Review of the manuscript Konopka, B. et al.: Influence of tree species and size on bark browsing by large wild herbivores
I consider the submitted manuscript (MS) an interesting contribution to studies regarding the the impact of ungulates on forest stands. Deep understanding of the factors (and their interactions) affecting the survival of the next generation is essential for the maintenance of sustainable management in forest stands. Moreover, the importance af such studies is highlighted by the fact that many of the Central European forests are currently facing the challenges of excessive density of the ungulates. Measures aiming at mitigation of negative effects of game represent one of the most important tasks for the management of forest stands.
The submitted manuscript is clearly structured and written, with sound methodology, up-to-date statistical analyses and a solid data-set. I appreciate also the extensive field work. After correcting few formal errors listed below as well as some shortcomings regarding the English, I recommend to accept this manuscript for publication in Plants.
Specific comments:
L57: is the largest herbivore
L58: consumes
L61: selects
L66: browsed
L74: are the most attractive
L100-106: this is just the repetition of values presented in Tab1, I recommend to write briefly about the differences bertween tree species
L105: omit one „in“
L129: significantly influenced
L130: delete „all p < 0.05“ - redundant
L132: delete „all p < 0.01“ - redundant
L135: delete „by large wild herbivores“ – redundant
L135: detto
Tabs2-4: correct the lines
In the captions of Figs2 and 3 use the italics for Latin names.
In Figs1-3 I recommend to move teh legends (for tree species) always to the first graph.
L170: replace „with“ by „in“
L172: delete occurred
L173: replace „was“ by „were“
L175: ...of the tree stem, bark browsing...
L182-183: differed between the years of browsing (Fig. 4).
L226: is
L228: ...to LWH as manifested by the largest...
L237: spruce-dominated
L238: replace „populations“ by „proportions“
L242: wood tissues, high on terpenoids concentration, especially botulin
L247: In study from Scotland
L255: repalce „to“ eith „in“
L288: than willow and rowan
L297: typo, correct to 3.2
L298: those of other studies
L310: unclear – why „focus“? Did you mean – „In more disturbed areas, these groupings could also contribute to better dispersion of the game across rhe stands“?
L312: dedicated plots – this sounds weird in English
L318: „The most practical solution“ – well, in my opinion, the mosty practical solution would be to keep the population densities on the level allowing sustainabkle forest management – briefly said – shooting
L3332: „which is about 5km from the town of Cadca“ – redundant information, not relevant
L339: replace „forest“ by beech
L340: replace „within“ by „during“
L348: delete „agricultural“, field is always argicultural
L349-352: deers
L353: wolf... are rare, however, their population density is slightly increasing in the entire region.
L358: delete „planting“, planting is always artificial
L364: replace „within“ by „inside“
L364: what do you mean by „supported growth“? did you mean „young growth“?
L377-382: in a 5-line sentence the rader easily gets lost – please divide into more sentences
L381: delete „by“
L384 10 year old
L385: replace „by“ by „of“
L389: I recommend to use the standard unit trees/ha, i.e. 28 000 trees/ha
L396: precision 0.01mm? well, such precision regarding stem diameter is an illusion and btw is of no ude here.
L412: delete „next“
L414-415: bark mass of a stem section was quantified as the product...
The entire Tab5 I recommend to move into Supplementary material
L428: „pre-browsed“
L443: delete „examine“
L459: replace „to“ by „in“
L465-466: As they are relatively short-living pioneers, what way do you recommend to keep them in a 100y-rotation (in general) commercial forests?
Author Response
We are really grateful for the valuable comments of the in helping us to improve the manuscript.
Hence, we incorporated almost all comments and suggestions in the manuscript, and provided rationale for any that we were unable to address.
Reviewer No1
I consider the submitted manuscript (MS) an interesting contribution to studies regarding the impact of ungulates on forest stands. Deep understanding of the factors (and their interactions) affecting the survival of the next generation is essential for the maintenance of sustainable management in forest stands. Moreover, the importance af such studies is highlighted by the fact that many of the Central European forests are currently facing the challenges of excessive density of the ungulates. Measures aiming at mitigation of negative effects of game represent one of the most important tasks for the management of forest stands.
The submitted manuscript is clearly structured and written, with sound methodology, up-to-date statistical analyses and a solid data-set. I appreciate also the extensive field work. After correcting few formal errors listed below as well as some shortcomings regarding the English, I recommend to accept this manuscript for publication in Plants.
Specific comments:
L57: is the largest herbivore – changed
L58: consumes– corrected
L61: selects– incorporated
L66: browsed – incorporated
L74: are the most attractive –- corrected
L100-106: this is just the repetition of values presented in Tab1, I recommend to write briefly about the differences between tree species – incorporated
L105: omit one „in“ – incorporated
L129: significantly influenced – incorporated
L130: delete „all p < 0.05“ – redundant – deleted
L132: delete „all p < 0.01“ – redundant – deleted
L135: delete „by large wild herbivores“ – redundant – deleted
L135: detto – deleted
Tabs2-4: correct the lines – deleted
In the captions of Figs2 and 3 use the italics for Latin names – incorporated
In Figs1-3 I recommend to move the legends (for tree species) always to the first graph
- modified
L170: replace „with“ by „in“ – incorporated
L172: delete occurred – incorporated
L173: replace „was“ by „were“ –corrected
L175: ...of the tree stem, bark browsing... – incorporated
L182-183: differed between the years of browsing (Fig. 4). – incorporated
L226: is – incorporated
L228: ...to LWH as manifested by the largest... – incorporated
L237: spruce-dominated – corrected
L238: replace „populations“ by „proportions“ – incorporated
L242: wood tissues, high on terpenoids concentration, especially botulin – incorporated
L247: In study from Scotland – changed
L255: repalce „to“ eith „in“ – incorporated
L288: than willow and rowan –corrected
L297: typo, correct to 3.2 – corrected
L298: those of other studies – incorporated
L310: unclear – why „focus“? Did you mean – „In more disturbed areas, these groupings could also contribute to better dispersion of the game across rhe stands“? - Yes, incorporated
L312: dedicated plots – this sounds weird in English - corrected
L318: „The most practical solution“ – well, in my opinion, the most practical solution would be to keep the population densities on the level allowing sustainable forest management – briefly said – shooting –agreement, incorporated
L3332: „which is about 5km from the town of Cadca“ – redundant information, not relevant - deleted
L339: replace „forest“ by beech – incorporated
L340: replace „within“ by „during“– incorporated
L348: delete „agricultural“, field is always agricultural - deleted
L349-352: deers – we think that it would be always “deer”
L353: wolf... are rare, however, their population density is slightly increasing in the entire region.
– incorporated
L358: delete „planting“, planting is always artificial - deleted
L364: replace „within“ by „inside“ – incorporated
L364: what do you mean by „supported growth“? did you mean „young growth“? – incorporated
L377-382: in a 5-line sentence the rader easily gets lost – please divide into more sentences
– OK, done
L381: delete „by“ - - deleted
L384 10 year old – incorporated
L385: replace „by“ by „of“ - done
L389: I recommend to use the standard unit trees/ha, i.e. 28 000 trees/ha – incorporated
L396: precision 0.01mm? well, such precision regarding stem diameter is an illusion and btw is of no use here. – incorporated
L412: delete „next“ - deleted
L414-415: bark mass of a stem section was quantified as the product... – incorporated
The entire Tab5 I recommend to move into Supplementary material - moved
L428: „pre-browsed“ - corrected
L443: delete „examine“ - deleted
L459: replace „to“ by „in“ - replaced
L465-466: As they are relatively short-living pioneers, what way do you recommend to keep them in a 100y-rotation (in general) commercial forests? – extra sentences were added about that.
Reviewer 2 Report
The authors present an interesting paper that is well written. The methods and results are strong, and the paper flows well. Please check the writing style as some sentences are too long (should not have sentences of more than 3-4 lines). Figures are blurry, should upload high quality figures. In figure legends, missing italics for species names. At line 235 you discuss data not presented, I suggest adding them as they are interesting to show, and you discuss them. Any reasons why they were not reported before and just discussed? Also, regarding the data analysis, why using Fisher LSD as post hoc test? LSD does not include any correction for multiple comparisons. Other tests such as Bonferroni-Holm are more appropriate. Missing information on which software was used for data analysis.
Author Response
We are really grateful for the valuable comments of the in helping us to improve the manuscript.
Hence, we incorporated almost all comments and suggestions in the manuscript, and provided rationale for any that we were unable to address.
Reviewer No2
The authors present an interesting paper that is well written. The methods and results are strong, and the paper flows well. Please check the writing style as some sentences are too long (should not have sentences of more than 3-4 lines).
English language was checked once again and improved by the coauthor - native speaker.
Figures are blurry, should upload high quality figures.
The Figures were replaced by ones of better quality.
In figure legends, missing italics for species names.
The species names are in the figure captions, perhaps there do not need repetition in the legends.
At line 235 you discuss data not presented, I suggest adding them as they are interesting to show, and you discuss them. Any reasons why they were not reported before and just discussed?
The exact results about browsing on Norway spruce are not really available, but the damage was negligible. Thus, we explained in the parenthesis that such data are not available.
Also, regarding the data analysis, why using Fisher LSD as post hoc test? LSD does not include any correction for multiple comparisons. Other tests such as Bonferroni-Holm are more appropriate. Missing information on which software was used for data analysis.
In fact, in the early version of the manuscript we had conducted both Fisher LSD and Bonferroni-Holm tests. Then, after comparing them, we decided for the Fisher LSD. The reason was that the LSD test were more conservative than Bonferroni-Holm tests, it gives more homogeneous groups (although differences were rather mild). Therefore, we would chose to use the LSD for our purposes.